# Geopolymer-Based Artificial Aggregates: A Review on Methods of Producing, Properties, and Improving Techniques

**DOI:** 10.3390/ma15165516

**Published:** 2022-08-11

**Authors:** Mohammad Almadani, Rafiza Abd Razak, Mohd Mustafa Al Bakri Abdullah, Rosnita Mohamed

**Affiliations:** 1Department of Civil Engineering, Faculty of Engineering—Rabigh Branch, King Abdulaziz University, Jeddah 21589, Saudi Arabia; 2Faculty of Civil Engineering Technology, Universiti Malaysia Perlis (UniMAP), Arau 02600, Perlis, Malaysia; 3Centre of Excellence Geopolymer and Green Technology (CEGeoGTech), Universiti Malaysia Perlis (UniMAP), Arau 02600, Perlis, Malaysia; 4Faculty of Chemical Engineering Technology, Universiti Malaysia Perlis (UniMAP), Arau 02600, Perlis, Malaysia

**Keywords:** artificial aggregate, geopolymer, sintering, cold bonding, autoclaving

## Abstract

The depletion of aggregate-related natural resources is the primary concern of all researchers globally. Recent studies emphasize the significance of recycling and reusing various types of natural or by-product material waste from industry as a result of the building industry’s rising demand for aggregate as the primary component in concrete production. It has been demonstrated that the geopolymer system has exceptional features, such as high strength, superior durability, and greater resistance to fire exposure, making it a viable alternative to ordinary Portland Cement (OPC) concrete. This study will examine the present method utilized to generate artificial aggregate-based geopolymers, including their physical and mechanical properties, as well as their characterization. The production process of geopolymer derived from synthetic aggregates will be highlighted. In conjunction with the bonding of aggregates and the cement matrix, the interfacial transition zone (ITZ) is highlighted in this work as an additional important property to be researched in the future. It will be discussed how to improve the properties of geopolymers based on artificial aggregates. It has been demonstrated that cold bonding provides superior qualities for artificial aggregate while conserving energy during production. The creation of ITZ has a significant impact on the bonding strength between artificial aggregates and the cement matrix. Additionally, improvement strategies demonstrate viable methods for enhancing the quality of manufactured aggregates. In addition, other recommendations are discussed in this study for future work.

## 1. Introduction

Natural crushed rocks or stones are commonly used as coarse aggregates in the production of concrete. Various types of natural rocks (e.g., limestone [1], basalt [2], and gravel [3]) have been employed in the concrete industry based on local geological circumstances. However, increasing construction demand and rapid consumption of rock resources have created a significant challenge to the sustainable development of the concrete industry. Concern for sustainable economic and social progress necessitates additional measures to alleviate the scarcity of natural aggregates and improve the sustainability of the concrete industry.

To address the aforementioned difficulties, three technological approaches have been proposed: recycling aggregates from waste concrete [4,5,6,7], direct utilization of coarse waste particles such as bottom ash and steel slag as aggregates [8,9], and generating artificial aggregates based on industrial wastes or byproducts [10,11]. Recycling aggregates involve the utilization of waste produced through construction and demolition (C&D) containing different types of materials such as steel, glass, and hazardous materials including asbestos and mercury, and for these different materials, further treatment is always required. Meanwhile, direct utilization of course waste such as bottom ash [8] and steel slag [9] also requires enhanced treatment such as wet treatment and steam pressure. In the meantime, towards focusing on sustainability and saving the environment, artificial aggregates, as an emerging technology, is a viable option for generating both fine and coarse aggregates, which can meet the demand for waste recycling and mass production. The potential of industrial wastes to be utilized as raw materials for producing artificial aggregates include coal fly ash [12], palm oil fly ash [13], and bottom ash [14].

In addition, for producing artificial aggregates with enhanced properties that are comparable to those of natural aggregates, the processing and producing methods should be focused on. Sintering [13,15] and cold bonding [16,17] are two well-known techniques for producing artificial aggregates. Sintering is heavily dependent on the application of a high temperature, which is often greater than 1000 °C for the crystallization of raw materials. In contrast, cementitious pastes are typically used in the conventional cold-bonding procedure to bind the raw components together. Compared to natural rock aggregates, these two types of artificial aggregates possess the properties of being lightweight, water-absorbent, and weak. However, since the production of sintered aggregates requires significantly more energy, cold-bonded aggregates produced at room temperature are considered more cost-effective and environmentally friendly for mass production, despite their lower strength, higher water absorption, and higher bulk density compared to sintered aggregates made from the same raw materials [11].

In order to improve the properties of cold-bonded aggregates, geopolymers can be introduced to achieve the target properties of artificial aggregates. According to a previous study, geopolymer lightweight aggregates (GLAs) do not require high temperature sintering since their strength is obtained through alkali activation of pozzolanic materials [18]. In comparison to the traditional cold-bonding technique using ordinary Portland cement binders, the geopolymerization method can be adjusted by a variety of factors, including precursor types [19] activator contents [20], activator types [21], and curing techniques [22], to achieve the desired properties.

Autoclaving technique is another method for producing artificial aggregates. For many years, autoclave-steam curing has been utilized to enhance the strength development of concrete products by using saturated vapor pressure. Due to the fact that the rate of hydration of cement increases with an increase in temperature, curing artificial aggregates under steam conditions can accelerate the gain in strength. In addition to temperature, the duration of steam curing is an essential factor in the development of compressive strength in artificial aggregates. As a result, sintering and autoclaving are the least desirable production methods due to their elevated temperature requirements, which ultimately increase the embodied energy and harmful emissions, hence compromising sustainability [23]. In contrast, cold bonding requires cementitious material and a longer curing time to obtain the requisite strength [12].

In order to limit the consumption of dwindling natural resources, it is necessary to extensively explore the issue of recycling waste materials and effectively utilizing artificial aggregates, particularly in the construction sector. Numerous studies on various natural and by-product materials as new artificial aggregates to replace natural aggregates have been reported. The incorporation of geopolymer technology has been shown to have the potential to improve the properties of new aggregates. Even though countries such as Brazil, Australia, and China have begun to employ recycled concrete aggregate (RCA) as a substitute to natural aggregate in concrete applications, the cost of upgrading the poor characteristics of RCA remains a major concern. As a result, recycling waste material to create new artificial aggregates is an effective alternative to replacing natural aggregate resources. Controlling the target properties of artificial aggregates can be accomplished by optimizing the mix design or including other materials such as geopolymers. Geopolymers can be used to create the desired properties of artificial aggregates. Additional approaches applicable for improving the properties of artificial aggregates include two-step pelletization, soaking in alkali solution, coating, and impregnation. Consequently, the purpose of this paper is to review the current research on the development of geopolymer-based artificial aggregates, including the processing and production of geopolymer-based artificial aggregates and aggregate performance-improvement techniques.

## 2. Geopolymers as Artificial Aggregates

In the construction field, specifically concrete production, it is well known that aggregates occupy more than 50% of the volume of concrete. However, the utilization and extraction of natural aggregates such as sand, stone, and gravel have been restricted in some countries or cities to preserve the environment due to the overexploitation of natural sources [22,23]. Therefore, some alternatives have been made towards minimizing the direct utilization of natural sources for aggregates as well as reducing environmental concern regarding increases in by-products with limited landfills available, which will consequently lead to pollution of the environment. Further initiatives to mitigate the depletion of natural resources include the introduction of recycled aggregates and artificial aggregates, which encourages the recycling of waste materials [15,24,25].

Recycled aggregates incorporate the direct utilization of industrial by-products as aggregates, such as masonry waste such as bricks or tiles, agricultural wastes, ferrous slag, and glass wastes. Generally, recycled aggregates consist of natural aggregates bonded with old cement mortars. This direct use of wastes as aggregates, however, has adverse implications on the mechanical and chemical properties of concrete, such as the leaching of hazardous elements into the concrete. In addition, despite extensive experiments, the usage of construction waste is also prohibited in some cases [26,27,28]. Despite efforts to control construction waste, the bulk of masonry waste such as concrete, bricks, and tiles, remains to be disposed of in landfills. Most construction concrete waste causes poor concrete performance in comparison to concrete made with natural aggregates [29,30], thus limiting its application in the construction field. As a result, research on increasing the qualities of recycled concrete aggregates is critical, as their utilization will have a favorable influence on both environmental and economic sustainability [31,32,33]. Moreover, towards improving the properties of the recycled aggregates, further treatment is always required as mentioned by Akhtar et al. [5]. The improvement technique includes separating recycled aggregates from small particles, dust, and weakly adhering mortar prior to application to the concrete for enhancing the mechanical and durability properties of the concrete produced [4], applying thermal and mechanical treatments to improve the properties of the recycled aggregates [6], and immobilizing microorganisms in recycled aggregates due to the conducive conditions of recycled aggregates for the survival of microorganisms [7].

Meanwhile, in order to reduce environmental concerns over growing waste products, artificial aggregates have been developed, with the manufacturing process still employing wastes as precursors. Artificial aggregates are environmentally friendly as they are primarily made from industrial waste. Recycling solid wastes into artificial aggregates contributes to sustainable development by decreasing the depletion of natural aggregates caused by the expansion of infrastructure [34,35,36]. In addition, artificial aggregate production with the inclusion of geopolymers has been described as having potential as a material with a strong surface layer and good structural performance, as well as being ideal to be used in lightweight concrete and other potential applications, including lightweight geotechnical fill and insulation goods. Some of the reported studies on geopolymer-based artificial aggregates are summarized in Table 1.

From Table 1, it can be noted that artificial aggregates are flexible alternatives to natural aggregates due to the variation in the producing methods starting from the shaping until the hardening of the aggregates via the curing process. These producing methods that include the mix parameters applied also lead to variation of aggregates properties achieved. For instance, a study reported by Huynh et al. [34] proved that the crushing strength properties of artificial aggregates made from fly ash and slag were enhanced by increasing the modulus silicate ratio (Ms) and were nearly comparable (about 89%) to the properties exhibited by natural aggregates.

Moreover, it was discovered that artificial aggregates generated from geopolymers and geopolymerization methods are viable alternatives to natural aggregates. As demonstrated in Table 1, fly ash is frequently utilized as a raw material in the production of artificial aggregates. This is attributed to the chemical composition of silicon (Si) and aluminium (Al) in fly ash, which plays an important role as a backbone in the geopolymerization process. Therefore, the use of fly ash channels the potential of geopolymers as artificial aggregates due to the simplicity of the production process and the minimization of waste production through the use of high-aluminosilicate materials. It is well known that alkali-activated materials (AAMs), which include geopolymers, have been introduced as a green material due to their favorable properties compared to ordinary Portland cement (OPC) and have a high potential towards reducing carbon dioxide (CO_2_) emissions as well as easy availability of precursors required [42,43,44]. The term alkali-activated materials (AAMs) is generally applied for any aluminosilicate materials reacting with high alkaline solutions. However, when utilizing aluminosilicate raw materials with little or no calcium such as fly ash (class F) with silicon and aluminum (Si + Al) as the main reactive binding agents, the term geopolymer is preferrable due to differences in chemical reaction processes [45,46]. Geopolymers are inorganic polymers that are typically produced by alkali activation (with an alkali activator) of aluminosilicate precursors (binding materials), as shown in Figure 1.

Geopolymers are suitable for use in construction field applications, including in artificial aggregate production as an alternative to natural aggregates, due to the simplicity of the manufacturing process via alkali activation, as well as having interesting and valuable physical, chemical, and mechanical properties comparable to those of ordinary Portland cement (OPC) [48,49,50]. The incorporation of geopolymers entails the use of the geopolymerization process during the creation of artificial aggregates as well as the use of crushed geopolymer paste according to the standard grade for application in concrete. Prior to utilization and processing, it is essential to identify the qualities of geopolymer-based artificial aggregates. The performance of the properties, including physical and mechanical properties, can be controlled by the precursors employed and the factors that determine the mix designation and mixing ratios.

### 2.1. Precursors

Common industrial by-products such as fly ash, palm oil waste, blast furnace slag, ferrochrome slag, and silica fumes have the potential to cause land shortages for trash disposal and boost the cost of waste treatment or disposal [13,51]. As a result, using these resources in concrete reduces waste generated by the respective industries as well as the carbon footprint of concrete manufacturing.

Waste material recycling and reuse have become increasingly significant to researchers in recent years, particularly in the geopolymer sector [46,52]. Geopolymers are well known to utilize materials that consist of high silica and alumina (usually >50% total sum of both SiO_2_ and Al_2_O_3_) as both of these compositions are significant as geopolymer backbones. Therefore, in order to utilize geopolymers as artificial aggregates, the selection of aluminosilicate precursors is crucial. Various aluminosilicate materials, such as kaolin, metakaolin, fly ash, slag, red mud, rice husk ash, and volcanic ash, have been extensively explored for their potential as geopolymer precursors, and the comparison of the chemical composition is shown in Table 2.

From Table 2, it can be concluded that there are varieties of raw materials that are applicable to be used as geopolymer precursors. Most of the materials used have high silicon (Si) and aluminum (Al), which are significant for geopolymer bonding as mentioned previously. The selection of raw materials is highly dependent on the application. For instance, kaolin- or metakaolin-based geopolymers are commonly applied for fire-resistant applications, and fly ash-based geopolymers are used as soil stabilizers and in construction applications such as artificial aggregates [15,59,60]. In addition, apart from the commonly used raw materials summarized in Table 2, other potential materials are also actively researched for utilization as geopolymer precursors such as granite [61], waste glass powder [62,63], and quarry tailings [41]. Therefore, in order to produce artificial aggregates from geopolymers, the selection of materials with high Si and Al that can be applied is significant in order to produce artificial aggregates that meet the requirement of properties or are comparable to those natural aggregates used in construction fields.

The use of geopolymers can be based on the Si/Al ratio, as the microstructure of geopolymers varies significantly depending on the Si/Al ratio. It is well known that Si and Al are the backbone elements of geopolymers. Therefore, an increased Si/Al ratio led to a denser microstructure of geopolymers produced, thus contributing to enhanced strength performance, and this was concluded by He et al. [64], which proved that a denser microstructure with increasing Si/Al ratio contributed to high mechanical property performance. The findings were also supported by Wang et al. [65], who determined the effects of Si/Al, Na/Al, and free water on micromorphology and the macro strength of metakaolin-based geopolymers. The characteristics of SiO_2_ and Al_2_O_3_, which are used to control the glass phase viscosity, are closely related to the properties of artificial aggregates, and according to Kwek et al. [66], aluminate plus silicate reactions are faster than silicate reactions alone. However, despite the use or recycling of industrial wastes for making artificial aggregates, the utilization of geopolymers as artificial aggregates has not been fully discovered, which is interesting to be explored towards producing environmentally friendly aggregates for construction fields. In addition, alongside factors controlling the geopolymerization for producing geopolymer-based artificial aggregates, the methods for production include shaping techniques as well as production or curing methods.

### 2.2. Influential Factors

Apart from the selection of precursors for geopolymer-based artificial aggregates, the composition ratio that controls the geopolymerization process is significant while preparing the mixture proportion of geopolymer-based artificial aggregates. These influential factors include the alkali solution concentration, alkali activator ratio (NaOH/KOH:Na_2_SiO_3_/K_2_Si_2_O_3_), and solid-to-liquid ratios (aluminosilicate materials and alkali activator) that are sometimes represented in molar ratios [67,68].

Both potassium hydroxide (KOH) and sodium hydroxide (NaOH) are often utilized as alkali solutions in the synthesis of geopolymers, generally in combination with K_2_Si_2_O_3_ or Na_2_SiO_3_ [69,70]. Alkali solution (MOH, M = Na^+^ or K^+^) is made up of alkali cations (Na^+^/K^+^) and hydroxide anions (OH^−^ ions). Both cations and anions in alkaline solution stabilize the negative charge of the aluminate (AlO_4_) tetrahedral structure, which enables the dissociation of bonds on the surface of raw materials [71,72]. The concentration of the alkali solution has a major effect on the characteristics of geopolymers, particularly during raw material dissolution to generate monomers. A slight modification of the molarity of NaOH utilized can significantly affect the characteristics of geopolymers, particularly their mechanical properties [73,74,75].

Meanwhile, apart from optimizing the molarity/concentration of the alkali solution, the ratio of the alkali activator should be addressed when sodium hydroxide and sodium silicate solution are used as an alkali activator [68,76]. Due to the enhanced mechanical properties of the resulting AAMs, the combination of sodium hydroxide (NaOH) and sodium silicate (Na_2_SiO_3_) is broadly applied as an alkali activator in geopolymers [76,77]. Since silica (SiO_2_) is more soluble than sodium (Na^+^), a sufficient amount of Na_2_SiO_3_ accelerates the nucleation and polymerization processes. The cation plays an important role in balancing the negatively charged tetrahedral aluminium sites in the polymeric network. Previously, it was found that increasing this influential factor led to an increase in the dissolution rate, as the Si element’s composition increased, which serves as a backbone for geopolymer bonding (Si-O-Si/Si-O-Al) [76].

Apart from controlling the composition of the liquid for geopolymer-based artificial aggregates, the solid-to-liquid ratio (S/L ratios) is also known as a significant parameter that controls the amount of solid precursor and liquid precursor for geopolymers. The S/L ratio will also have a significant effect on the chemical composition of the geopolymerization process. Generally, increasing the S/L ratio (solid composition) accelerates geopolymerization due to the availability of Si and Al elements required for geopolymer bonding. A favorable S/L ratio is necessary to ensure effective geopolymerization and hence the production of geopolymer-based artificial aggregates with better performance is achieved [71,78]. Meanwhile, a further increase in the S/L ratio leads to insufficient liquid for reaction, causing improper bonds to be formed [79].

In a nutshell, these influential factors are widely reported in order to optimize the mix designation for better properties of the geopolymer-based artificial aggregates produced. In order to produce artificial aggregates from geopolymers, apart from the selection of aluminosilicate precursors and the effect of influential factors, the technique and methods for processing and producing the artificial aggregates are crucial. This is because geopolymerization requires the use of sodium hydroxide, which has been identified as hazardous to the environment if used excessively. As a result, optimization of each of the influential factors, as well as materials utilized as precursors due to heavy metal leaching of the materials derived from industrial wastes, should be stressed and investigated further in the future.

## 3. Processing and Producing Geopolymer-Based Artificial Aggregates

Generally, geopolymer-based artificial aggregates are produced via the geopolymerization process that involves three steps, which are the dissolution of Si and Al elements from raw materials, transportation or orientation of precursor ions into monomers, and polycondensation of monomers into polymeric structures. In order to produce artificial aggregates, after mixing of both solid and liquid precursors with the respective mix designs, shaping of the aggregates is required. There are various methods for shaping the aggregates, including pelletization, mold casting, crushing, and hand shaping [80].

The pelletization process is commonly used to manufacture pelletized aggregates where several considerations should be made for the efficiency of the production of pellets, including the speed of revolution of the pelletizer disc, moisture content, angle of the pelletizer disc, and duration of pelletization [81]. The pelletization process also includes an agglomeration step as well as a granulation step. The granulating procedure affects the properties of aggregates due to the binding forces, including (i) the interaction force and surface tension of liquid, (ii) the viscosity/adhesion of the binder, (iii) the adhesion of the electric double layer, and (iv) the Van der Waals force and electric charge attraction between powders [82]. The first two are the main factors controlling granulation. Granulating qualities are controlled by the granulating method, granulator operating conditions, catalysts, and the properties of binders or additives added. In the meantime, it has been recognized that mold casting is more ideal for laboratory-scale research. The crushing process is also widely utilized because it has been proven to be effective for crushing natural aggregates into the desired grades. Hand shaping is considered a traditional technique because it requires human labor and cannot be scaled up for mass manufacturing.

Apart from shaping the freshly created aggregates, the curing processes for manufacturing geopolymer-based artificial aggregates must be taken into account. Artificial aggregates can be produced by converting various binder materials and manufacturing procedures such as pelletization followed by a curing and hardening process involving cold bonding, sintering, and autoclaving [83], summarized in Table 3.

### 3.1. Curing Method

#### 3.1.1. Cold Bonding

Cold bonding is a method of hardening artificial aggregates by air drying them in an enclosed environment for one, three, or seven days at a low temperature (below 100 °C). Cold-bonded aggregates are often cured with water or covered with plastic sheets until sufficient strength is obtained. Cold bonding is known as an energy-saving approach since it does not require any additional heat during the process [15] and it is widely used when utilizing cementitious materials such as coal fly ash, iron ore tailings, and granulated blast furnace slag, as the key to aggregate strength gain is through a cement hydration and/or pozzolanic reaction [85]. In addition, according to Gesoglu et al. [86], the use of fly ash lightweight aggregates manufactured by the cold-bonding method leads to lower energy consumption and greater cost-effectiveness compared to sintering.

A study reported by Bui et al. [82] utilized cold bonding during the production of fly ash-derived artificial aggregates. As additives, ground granulated blast furnace slag (GGBS) and rice husk ash (RHA) were added to the fly ash to improve the aggregates’ properties. The addition of GGBS and RHA to the fly ash led to increases in calcium oxide (CaO) and silicon oxide and aluminium oxide (SiO_2_ and Al_2_O_3_), which are reactive during the alkali activation process. The effect of the addition of GGBS and RHA was observed on the strength properties as well as water absorption of the aggregates produced. Granulation was used to produce the aggregates, and the activator solution was sprayed onto the fly ash in accordance with the mix design (FA-GGBS, FA-RHA, and FA-GGBS-RHA). The optimal alkali activator spray was chosen based on the mix design ratios. According to the results, increasing the GGBS composition resulted in the highest increase in the crushing strength performance (15.5–15.7 MPa) compared to the addition of RHA (6.0–8.1 MPa) and the addition of both GGBS and RHA blended with fly ash (8.1–8.8 MPa). This can be explained by the calcium content, which causes the production of calcium–silicate hydrates, which contribute to the aggregates’ strength development. This occurrence was also supported by the lowest percentage of water absorption when utilizing blends of FA-GGBS (7.8–8.3%). From the study, it can be concluded that additives with high calcium are significant when applying cold bonding in order to gain better properties without any additional energy supply.

A study reported by Risdanareni et al. [84] attempted to solely utilize fly ash as a raw material for aggregate manufacturing using the cold bonding method towards minimizing the energy consumption. Despite the use of a high calcium binder to enhance the aggregates produced, this study determined the effect of the concentration of NaOH on the properties of aggregates produced, which is commonly known to affect the properties of geopolymers. By using an agglomeration technique for producing the aggregates, the alkali activator used, NaOH at different concentrations (4 M, 6 M, 8 M), was sprayed for 10 min onto fly ash in the granulator pan with a diameter of 500 mm. This led to the formation of granules during the pelletization process. The granules formed were allowed to cure at room temperature of 20 °C for 24 h. The dried granules or noted as the aggregates were then sieved and stored for 28 days prior to utilization. From the results reported, it was found that increasing the concentration of NaOH led to increasing density, thus reducing the water absorption of the aggregates. The increasing concentration of NaOH also showed the decreasing intruded volume during porosity determination, thus proving the reduction of pore formation, which was consequently proved by the high early strength performance gained by the aggregates.

Another study was carried out by only utilizing commonly used materials as raw materials for geopolymers with the addition of ordinary Portland cement (OPC) and calcium hydroxide (Ca(OH)_2_). Instead of using an alkali activator, this study emphasized the utilization of water as a binder solution and cold bonding during the manufacturing process of artificial aggregates. The effect of OPC addition (5–15%) and Ca(OH)_2_ to fly ash was observed through density and water absorption of the aggregates produced. The pelletization was carried out using a disc pelletizer followed by cold bonding by curing at ambient temperature for 28 days prior to utilization. It was found that the density of the artificial aggregates produced was enhanced with increasing both OPC and Ca(OH)_2_ (1765 kg/m^3^ and 1737 kg/m^3^, respectively) compared to the fly ash aggregates (1728 kg/m^3^). However, too high an increase in the percentage addition of both OPC and Ca(OH)_2_ reduced the density, and this was explained due to the improper reaction process with limited liquid available. This also explained the increasing water absorption percentage when the highest amounts of OPC and Ca(OH)_2_ were added to the fly ash. From this study, it can be concluded that in order to improve the proper reaction process for the formation of the aggregates, a geopolymerization process can be adopted during the production that involves the utilization of an alkali activator. By using the proper mix designation by controlling the influential factors and precursors applied, the geopolymerization has potential to produce aggregates with better performance of physical and mechanical properties.

#### 3.1.2. Sintering

Sintered aggregates are created by fusing particles of fresh pellets at a high temperature (often above 1000 °C). The aggregates are gradually created at high temperatures by the expansion and vitrification of fresh pellets. Sintering is a somewhat complex process involving physical and chemical reactions; hence, the properties of sintered artificial aggregates are mostly determined by the raw material characteristics and sintering parameters. The sintering parameters include the sintering temperature effect as well as sintering duration. Similar to the manufacturing process of cold bonding, the fundamental production process of sintered artificial aggregates consists of three stages; mixing process of raw materials, pelletization or granulation process, and hardening process through sintering at high temperature [87,88]. Regardless of energy consumption, sintering is commonly denoted as a time-saving hardening process since the duration of sintering usually takes around 1 h for pre-heating and the sintering process.

Generally, the sintering process is known as one of the reliable methods to produce high-quality, lightweight artificial aggregates in terms of the mechanical properties of the aggregates that are comparable to the standard commercialized lightweight aggregates such as Leca and Lytag. Due to the enhancement of the reaction process with additional heat supply, sintering is commonly used for producing artificial aggregates, specifically lightweight artificial aggregates. With the additional energy supply, the sintering process is applicable and has potential to be applied for variation of raw materials and is not limited to materials with cementitious properties only.

Numerous studies have been carried out in order to evaluate the effectiveness of the sintering process towards aggregates produced as tabulated in Table 3. For instance, Kwek et al. [13] carried out a sintering process for the production of artificial aggregates made from palm oil fly ash (POFA). During the sintering process, preheating at 400 °C was applied prior to sintering at 1150 °C in order to avoid sudden expansion, which leads to cracking due to the high heat supply, thus negatively affecting the properties of the aggregates formed. The artificial aggregates were noted as capable for lightweight application due to the results obtained, such as the density, which was within the range of standard-grade lightweight properties. This study also proved the potential of POFA as a raw material for producing artificial aggregates due to the significant decrease in water absorption with increasing POFA content. This study proved that the sintering process is capable of developing stronger bonds with the additional heat supply, thus helping to improve the properties of the aggregates produced.

The utilization of raw materials other than fly ash was also carried out by Balapour et al. [14], who reported the sintering process during the manufacturing of high calcium and low calcium coal bottom ash aggregates towards evaluating the potential of the aggregates for internal curing in concrete. A sintering temperature of 1150 °C was used in this study. The study concluded that the calcium content played a significant role in the development of pores in the artificial aggregates, thus suggesting the difference in water absorption percentage achieved. However, regardless of different amounts of calcium, artificial aggregates with NaOH presented as an activator solution showed some promising properties for internal curing in concrete due to the excellent properties obtained including adsorption/desorption, water absorption, and porosity, which were within the standard.

A similar observation was found by Terzic et al. [15], who observed the potential of fly ash aggregates made from cold bonding and sintering processes. A sintering temperature of 1100 °C was applied in the study. The comparison of properties was observed based on different curing regimes and from the results reported, it was found that sintered artificial aggregates showed a denser structure with smaller non-interconnected pores, suggesting less water absorption compared to cold-bonded artificial aggregates, which were open and interconnected. These occurrences also affected the properties of the concrete produced. Concrete with cold-bonded aggregates had higher porosity compared to sintered-aggregates. This proved the potential of sintering as a promising curing method for producing concrete with a better strength performance.

Incorporating geopolymer also adds to homogeneous pore distribution and allows for the production of lightweight aggregates at lower temperatures [88]. Thus, it is considered that artificial aggregates with a geopolymer comprising volcanic ash as the principal component improve the qualities of aggregate produced at temperatures below 1000 °C. This study showed that the incorporation of geopolymers can save energy during the sintering process, where low sintering temperatures are used to make artificial aggregates, compared to the absence of geopolymer incorporation. Further research should focus on the performance of geopolymer-based artificial aggregates for various materials, such as Si and Al precursors, and compare this to the performance of aggregates without geopolymer inclusion.

#### 3.1.3. Autoclaving

Autoclaving is a process that requires the addition of chemicals, such as lime or gypsum, during the agglomeration phase. Autoclaving leads to the production of artificial aggregates with minimum binding material with a short curing time [1,72]. The commercialized autoclaved building products are usually prepared with calcareous materials including cement and quicklime, siliceous materials such as quartz sand or fly ash, and with H_2_O under pressurized steam at 125–200 °C, giving an insight into the potential of producing artificial aggregates using autoclaving. During the production of aggregates, the autoclave’s pressure and temperature will reinforce the pellets produced by pelletization. Up to this date, there are limited studies available on exploring the potential of using autoclaving during the manufacturing of artificial aggregates, specifically involving geopolymers.

A study reported by Wang et al. [19] utilized quartz tailings (QT) as raw material for producing artificial aggregates (QTAs). Quartz tailings were mixed with fly ash as well as quicklime in order to ensure the homogeneity of the chemical composition of the raw materials. Quicklime was used in this study in which the composition was controlled by using CaO/SiO_2_ ratios. The raw materials were then combined with 20–25 wt% water and sealed for 4 h to digest quicklime. The mixtures were poured into a self-designed pelletization disc to generate 5–16 mm QTAs. Raw QTAs were cured for 24 h naturally and then autoclaved. In the first three hours, the curing temperature rose from 20 °C at ambient temperature to 195 °C at an autoclave pressure of 1.38 MPa. The temperature was then maintained at 195 °C for another 10 h until reaching room temperature. With the promising physical properties including less water absorption (7–21%) and density (1008–1087 kg/m^3^) that are in line with standard lightweight aggregates, the QTA in this study was noted as having potential to be used as an artificial aggregate, specifically in lightweight application. This was also proven by the comparison of the high compressive strength of concrete obtained when utilizing QTA (74 MPa) and concrete with natural aggregates. This shows that quarry tailings are worthwhile to be explored as precursors for producing artificial aggregates alongside the positive effect on the environment due to minimizing the tailings produced.

Another study by Wang et al. [41] was also carried out by utilizing quarry tailings. According to the study, quarry tailings are typically stockpiled due to their stable crystalline structures below 100 °C and abundance, resulting in major environmental damages and considerable ecological risks. According to previous research of Wang et al. [19], the enhancement of reactivity of tailings can be achieved via autoclave technology, which is worthwhile to be explored. In this study, the influential factors were more centered on the curing conditions, including (1) investigating the effect of curing temperature on the properties of LWAs from 25 to 190 °C (25, 130, 150, 170, and 190 °C) at a saturated steam pressure (1.25 MPa) and curing time (4 h), (2) varying the steam pressure (0.50, 0.75, 1.00, and 1.25 MPa) at a constant temperature (150 °C) and curing time (4 h), and (3) increasing the curing temperature (190 °C) while decreasing the cement content (from 30 to 10 wt.%) without increasing the curing time. By controlling the curing conditions, the properties of artificial aggregates were found to be significantly affected. With an increase in curing temperature from 25 to 190 °C, the strength of LWAs increased from 2.48 to 11.95 MPa and water absorption decreased from 11.2% to 2.09%, indicating high potential in civil engineering applications. From both of these findings, it can be concluded that the autoclaving method can be further explored for other materials utilized as precursors for aggregates due to the promising properties reported.

In conclusion, processing and producing geopolymer-based artificial aggregates involve three steps, which are mixing, pelletization (sometimes denoted as agglomeration and granulation), and curing. From Table 3, various materials can be applied for producing geopolymer-based artificial aggregates. However, the performance of geopolymer-based artificial aggregates was found to be enhanced when utilizing an alkali activator (combination of NaOH and Na_2_SiO_3_), which is a liquid precursor of the geopolymerization process compared to utilizing water as liquid for the production of the aggregates, thus suggesting the potential of geopolymers and geopolymerization in artificial aggregate production.

In terms of curing, cold bonding, sintering, and autoclaving are noted as efficient methods to be applied for producing geopolymer-based artificial aggregates. Cold bonding is noted as an energy-method due to no additional heat supply or energy addition during the curing process. This method is commonly applied when utilizing high calcium materials due to fast development and hardening of the aggregates, and the curing process occurs naturally over time. Sintering involves additional heat supply, which is significant for enhancing the early age properties of the artificial aggregates. This process is suitable when utilizing materials with slow setting, such as clay-based materials. The development of aggregate bonding and the hardening process can be enhanced via additional heat supply but require optimal temperature of sintering to avoid cracking the outer shell of the aggregates. Autoclaving is known as an effective method with additional heat or pressure applied. However, regardless of its reliability and fast curing process of aggregates, this method is rarely reported by previous studies due to further equipment (autoclaving machine) required, which is considered as not efficient in terms of cost.

## 4. Properties of Geopolymer-Based Artificial Aggregates

The determination of properties is significant in order to observe the potential of artificial aggregates made in line with the standard properties of the natural aggregates. There are numerous properties that have been determined while evaluating the potential of artificial aggregates including the appearance, which covers the shape and size of the aggregates produced as well as the physical properties, mechanical properties, and morphology of the aggregates. Among all of these properties, the physical and mechanical properties of the artificial aggregates are the most widely reported for reflecting the performance of the artificial aggregates produced. Some of the reported studies on physical and mechanical properties are summarized in Table 4.

### 4.1. Physical Properties

The physical properties that are significant to be evaluated include specific gravity, density, and water absorption. The physical properties of artificial aggregates can be evaluated at an early age of 7 days or 28 days after curing prior to utilization in concrete production. Table 4 presents some reported studies on the specific gravity, density as well as water absorption of geopolymer-based artificial aggregates. Generally, regardless of similar materials used for the production of geopolymer-based artificial aggregates, it can be seen that the results for almost all of the physical properties reported including the specific gravity, density, and water absorption showed an irregular or random pattern of results. This occurrence proved that the performance of aggregates was significantly affected by the influential factors applied as well as the processing method. From Table 4, most of the reported studies on artificial aggregates associated with geopolymers and geopolymerization had a specific gravity in the range of 1.6–2.5 due to the processing of the aggregates. For instance, as reported in Table 4, the addition of ground granulated blast furnace slag (GGBS) to the fly ash for producing aggregates leads to higher specific gravity compared to the addition of silica fume to the fly ash, which could be due to the utilization of calcium during the geopolymerization process [40,41,86,87].

In addition, in terms of method processing, as reported in Table 4, it was found that sintered artificial aggregates had lower specific gravity (less than 2.0) of aggregates produced compared to cold-bonded artificial aggregates, suggesting the potential of artificial aggregates derived from geopolymers for lightweight application, as according to BS EN 13055-1, the specific gravity of lightweight aggregates should be less than 2.0. This is due to the reduction of weight loss by the firing of unburnt coal and the formation of internal gases during the sintering process, thus creating voids and pores [80]. For cold-bonded aggregates, the addition of foaming agent is crucial for utilization in lightweight applications.

Due to the correlation between density and specific gravity, previous studies typically reported either of these properties. In addition, the previous literature that measured the density of artificial aggregates reported bulk density. Similarly, for specific gravity, the density of geopolymers derived from artificial aggregates varied greatly depending on the applied influential factors, precursors, and processing method. The potential of artificial aggregates in lightweight application was also proven by the bulk density reported in previous studies as outlined in Table 4. According to Table 4, the bulk density of most reported research was less than 1500 kg/m^3^, which is comparable to the density of normal aggregates yet better for lightweight application [12,92].

Apart from specific gravity and density, other physical properties commonly reported and noted as significant for artificial aggregates include water absorption. The absorption of water indicates the interior aggregate structure. Higher water absorption of aggregates suggests a significant number of pores in nature, which usually results in disadvantages for the aggregates, and this was highly affected by the processing method of the aggregates. According to Table 4, most of the studies presented a tolerable percentage of water absorption (less than 25%), and it was also found that most of the low density materials achieved had a higher percentage of water absorption. This occurrence was due to more capillary pores available for absorption, thus causing increased water absorption, specifically at the early stage of seven days [12]. Rehman et al. [12] also compared the percentage of water absorption between artificial aggregates derived from geopolymers and artificial aggregates derived from cement and reported that artificial aggregates derived from geopolymers had higher water absorption compared to cement-based artificial aggregates due to the hydration process of cement. This occurrence showed that the precursors involved in processing artificial aggregates are significant, and this was also supported by Sharath et al. [90] and Gomathi et al. [89], who determined the effect of the addition of materials to the fly ash for producing geopolymer-based artificial aggregates.

### 4.2. Mechanical Properties

Apart from physical properties, the performance of geopolymer-based artificial aggregates is commonly evaluated through mechanical properties. The mechanical properties include the percentage of the aggregate impact value (AIV) and the percentage aggregate crushing value (ACV) as well as the crushing strength. The percentage AIV represents the resistance of aggregate to failure by the impact load. The percentage ACV is known as the percentage by weight of crushed (or finer) material obtained when test aggregates are subjected to a particular load under standardized conditions. In addition, the physical properties are correlated with the mechanical properties produced by the artificial aggregates. For instance, lower density or higher water absorption leads to a higher AIV percentage of aggregates. From Table 4, it can be seen that higher water absorption (more than 5%) commonly leads to a higher AIV value or ACV value.

In addition, according to Rehman et al. [12], the AIV value of artificial geopolymer aggregates was reduced significantly (39.00% to 25.00%) with increasing addition of GGBS to the fly ash and displayed AIV within an acceptable range according to BS-812–12 after 28 days of curing. This was due to the additional C-S-H and calcium-based geopolymer formation causing a denser microstructure [12]. This proved that the alteration of the chemical composition of the geopolymer-based artificial aggregate precursors significantly affected the mechanical properties of the aggregates produced. According to the literature, a pellet strength of 4.72 MPa was produced when only fly ash was used in the production of AAs, but it reached 22.81 MPa when GGBS was added [89]. On the other hand, the curing method also plays a significant role in the mechanical properties of artificial aggregates produced. Cold-bonded artificial aggregates have lower crushing strength compared to sintered artificial aggregates, which is related to the diameter of the aggregates [87].

### 4.3. Morphological Properties

The morphological properties of artificial aggregates were determined by previous researchers for observation of the pores of aggregates produced, which is significant for further observation on the results of density and water absorption [39,90,92]. For instance, Kwek et al. [13] compared artificial aggregates made from palm oil fly ash (POFA) and alkali activators of NaOH and Na_2_SiO_3_. The morphology, as depicted in Figure 2, showed the comparison of the aggregates before sintering and after sintering. From the morphology, it was found that sintered aggregates had a porous structure with homogeneous and coherent morphology compared to before sintering. Crystallization and vitrification were observed when the sintering temperature reached 1150 °C. According to the study, the pore diameter expanded during the sintering process, the vitrified layer melted, and blown-off gas bubbles occurred in the interior of the aggregate samples, thus resulting in a microstructure that was internally porous and dense.

Tian et al. [37] also reported on the morphology of its cold-bonded artificial aggregates derived from red mud and fly ash in which the correlation of the mix design was determined and correlated with the microstructure formation and the densities of the aggregates produced. Decreasing the composition of red mud in the fly ash was believed to cause the internal pores to change from small and regular to large and irregular, and the compactness of the internal structure started to decrease as the ratio of RM to FA decreased, which was consistent with the change in the densities of the aggregates and the conclusions obtained.

In addition, morphology is noted as one of significant properties for observing the insight of aggregates formation as being depicted by past researches [93,94]. Morphology determination is also crucial during application of the aggregates in concrete, specifically in the interfacial transition zone (ITZ), which is denoted as the region of the boundary observed when applying aggregates to concrete.

## 5. Interfacial Transition Zone (ITZ)

Conventional concrete has a region between the cement paste and the aggregate known as the interfacial transition zone (ITZ), which is regarded as the most crucial interface. According to Ollivier et al. (1995) [95], during the casting process of fresh concrete, a gradient in the water-to-cement ratio forms around the aggregate particles, changing the microstructure of the hydrated cement paste in the area. The interfacial transition zone (ITZ) is the area surrounding the aggregates.

A narrow ITZ (half of the conventional ITZ thickness) and high microhardness values in the artificial aggregate samples showed an improvement in ITZ. It can be explained by the continued reaction of the outer layer of artificial aggregate with the preexisting cement matrix, which results in less porosity and orientation deposition of crystals [96]. The strength of artificial lightweight concrete increased as a solid cement matrix was produced by creating a C-S-H gel around the artificial aggregate, which increased the density of the ITZ as being reported by Kwek et al. [13] in which is depicted as in Figure 3. Moreover, in other study, the geopolymer aggregate/matrix ITZ appeared to be wider (about 7 µm), and the microhardness value increased as the geopolymer aggregate successfully formed a strong chemical connection with the cementitious matrix and might be regarded as “additional defects” in the high-strength matrix [97].

Huang et al. (2019) [98] also claimed that there were no obvious cracks at the ITZ junction due to the tight binding between the lightweight aggregate and cement paste. This encouraged mechanical interlocking conditions and cement hydration products to deposit in surface flaws where pozzolanic reaction products were less likely to contribute. Furthermore, the microhardness and resistivity of the sintered fly ash aggregate supplied by the ITZ were higher than those of the granite aggregates. As the specimen with sintered fly ash aggregate was more resistant than the specimens with paste alone and the granite aggregate, the sintered fly ash aggregate concrete had lower porosity in the ITZ than the granite aggregate concrete. Furthermore, it has been demonstrated that extending the curing periods for both aggregates reduces the porosity of the ITZ [99]. Additionally, high ITZ thicknesses and dense ettringite and portlandite deposits were visible over the lightweight concrete in the early eras. An increase in the cement dose and a drop in the water-to-cement ratio generated C-S-H production. ITZ thicknesses decreased as the lightweight concrete compressive strength increased [100]. In addition, the geopolymer aggregate concrete generated a fracture network due to the presence of a greater number of microcracks in the cement-geopolymer aggregate transition zone than at the cement-natural aggregate interface at an early age. Similar to the natural aggregate concrete, the geopolymer aggregate concrete still exhibited a well-compacted and denser ITZ after 28 days. At 28 days, the greater density of the ITZ increased the aggregate-matrix bond, increasing the compressive and flexural strengths of the geopolymer aggregate concrete [101].

In comparison to natural aggregate, artificial aggregate using geopolymer has fewer microcracks. According to energy-dispersive X-ray (EDX) research, the hydrates were largely a C-S-H gel in the form of granular whisker-like hydrates and flake-like crystals in the concrete formed of geopolymer aggregates. Furthermore, the ITZ of natural aggregate concrete indicated that a water film often forms around the coarse particles due to bleeding and wetting effects. This results in a more porous microstructure in the ITZ, which results in an increased water content. As a result, the concrete with geopolymer-based artificial aggregate did not show weak ITZs in the area of the aggregates, although conventional concrete did with voids. Furthermore, geopolymer ITZ has higher mechanical interlocking forms due to the penetration of cement paste in geopolymer-based artificial aggregate, which results in strong aggregate-to-paste adhesion [50]. In addition to contributing to the performance of concrete, the pozzolanic reaction between aggregate and the presence of CaOH within the ITZ also plays a role. In geopolymer concrete, ITZs with high microhardness are connected to a layer of paste with dense microstructures and a high gel content. As a result, ITZ for artificial aggregate-based geopolymer exhibited better improvement in properties when compared to natural aggregate utilized in concrete production.

## 6. Improvement Techniques for Artificial Aggregate-Based Geopolymer Properties

Most artificial aggregates are porous and contain air voids, which lead to their classification as lightweight aggregates [102]. Due to their internal porous structure, artificial aggregates have a higher water absorption rate than natural aggregates [91,100,103]. Furthermore, artificial lightweight aggregates are highly porous and spherical, with densities less than 1200 kg/m^3^. Water absorption ranges from 5 to 20% by weight and is due to the dispersion of non-spherical pores that do not follow a predictable pattern [104]. Water absorption is somehow higher than acceptable ACI values because of the porous internal structure of the geopolymer-based artificial aggregate [105]. Due to the porous nature of artificial aggregates, it is necessary to treat aggregates in order to enhance their properties. To improve the qualities of artificial aggregates, two-step pelletization, alkali solution soaking, coating, and vacuum impregnation are used, as detailed in Table 5.

### 6.1. Two-Step Pelletization

The double-step pelletization process improves the physical, mechanical, and stabilizing properties of the binding matrix. The reaction process can be illustrated as in Figure 4.This approach produces an encapsulating outer layer made of a waste-free binder, which lowers porosity, prevents heavy metal release, and improves the mixture’s physical and mechanical properties [101]. Furthermore, expanded perlite particles can be contained within a shell structure utilizing fly ash to produce core-shell structured lightweight aggregates with particle densities ranging from 0.88 to 1.14 g/cm^3^ and bulk crushing strengths ranging from 2.04 to 2.66 MPa [108]. In addition, the two-step pelletization process requires 80% waste concrete powder (WCP) and 20% phase change material for the lightweight aggregate (PCM). The resulting core-shell PCM aggregate has a good capacity for storing energy and adequate strength [107]. Two-step pelletization will create a physical barrier that help to prevent the leaching of material in the aggregates. According to Yang et al. (2021) [106], the compact shell is the first barrier water molecules must penetrate to reach the core during leaching. The shell serves as a physical barrier that slows the movement of water molecules and the solubilization of chromium. A thick protective layer may trap chromium contamination in the core and prevent it from escaping, hence reducing the leachability of chromium.

From Table 5, the lightweight aggregate produced did not use geopolymer as a binder in the production rate. The fly ash and cement were the common materials used for treatment in the two-step pelletization. In comparison to artificial aggregate without an extra outer layer, this method allowed for an improvement in the physico–mechanical characteristics and a decrease in heavy metal leaching. However, the artificial aggregate-based geopolymer has not been studied for the treatment of two-step pelletization, and this is required for further investigation by using geopolymer as a treatment material.

### 6.2. Soaking in Alkali Solution

The soaking of artificial aggregate is a novel technique for enhancing aggregate properties, as it increases the matrix’s strength through geopolymerization and can be illustrated as in Figure 5. To improve the quality of the ITZ between the artificial aggregate and the geopolymer matrix [106], it is advised that coarse aggregate be soaked in a NaOH solution [109]. Compared to water-cured aggregates, the strength of the fly ash aggregates that were chemically cured by immersion in a sodium hydroxide (NaOH) solution generated with various molar ratios was found to be 64.93% higher after 7 days and 49.03% higher after 28 days [110]. The surface treatment of cold-bonded fly ash aggregate by immersing it in sodium silicate solution significantly improved the aggregate strength and water absorption [111]. In a nutshell, soaking artificial aggregate in an alkaline solution for a minimum of one hour will increase the aggregate properties, hence enhancing the durability and mechanical properties of the concrete when apply.

From Table 5, studies on soaking lightweight aggregate in alkaline solution is very limited. The lightweight aggregate is soaked either in sodium hydroxide or water glass, which is type of alkaline solution. The lightweight aggregate produced from fly ash is usually used to carry out the treatment to improve the properties of the aggregates. Since the treatment material is sodium hydroxide, an alkaline solution used to produce geopolymer, soaking artificial aggregate in an alkaline activator consisting of sodium silicate and sodium hydroxide solution is believed to improve the properties of artificial aggregates. However, the process of coating can be accentuated by the effect of coating thickness and the appropriate viscosity of the solution. Future research should focus on the relative humidity and time required for soaking or immersion in an alkali solution. Therefore, the artificial aggregate-based geopolymer that soaked in an alkaline activator requires further investigation in order to determine whether the geopolymerization process will help to improve the properties of artificial aggregates.

### 6.3. Coating

The civil engineering sector has seen a significant increase in demand for both environmentally friendly and sustainable coatings that are applied to artificial surfaces with better performance. The stronger and significantly more impermeable shell structure produced by the coating process, as evidenced by the higher bulk crushing resistance and lower water absorption, has excellent efficiency in improving the properties of artificial aggregates [11]. Coating method, as being depicted in Figure 6, require lesser time of 15–30 min for the coating process compared with soaking in alkali solution. When evaluating the strength and water absorption of coated artificial aggregates, the OPC-silica fume mix coating performed marginally better than OPC-fly ash coatings [112]. In addition, the artificial aggregates might be further improved and protected from water damage by being coated with polypropylene, a type of plastic that reduces the specific gravity and water absorption while boosting particle strength. The artificial aggregate with a shell-core structure created by plastic coating improved the compressive strength of concrete and the thermal insulation of concrete compared to concrete prepared with normal aggregates [59]. In addition, the filling of aggregate pores by phase change material through submersion and the influence of the coating material by silica fume and cement will improve the specific gravity while lowering absorption [113].

The treatment for lightweight aggregate-included geopolymers is very limited. The lightweight aggregate using geopolymer has been coated with plastic, showing an improvement in the properties of the lightweight aggregates. The artificial aggregate-based geopolymer has proved that the optimum strengthening effect that plastic coating technology applied to it allows geopolymers to be utilized in more profitable markets such as sealing nuclear waste and heavy metals in the future. A porous aggregate is selected as a coating as it can fill up the pores around the surface of lightweight aggregates. However, knowledge of the stiff state of covering with geopolymer as a treatment is limited. The optimal viscosity of the solution slurry should be stressed for varied materials. Future studies must also investigate the interaction between water absorption and the soaking rate in the geopolymer slurry. Thus, there is lack of studies on the coating of lightweight aggregates with geopolymer inclusion, requiring further investigation to determine its benefit to the environment.

### 6.4. Vacuum Impregnation

Vacuum impregnation is a new novelty method used to improve the properties of lightweight aggregate by removing the air trapped in the aggregate’s pores and filling it up with materials such as phase change material. As being illustrated by Figure 7, the lightweight aggregate will undergo vacuum impregnation with paraffin oil and two-layer coating, with epoxy resin as the first layer and silica fume as the second, which both exhibit outstanding thermal performance [114]. Expanded perlite mixed with paraffin and covered with polystyrene has shown to offer considerable benefits for thermal energy storage in concrete because of the high latent heat of paraffin and the exceptional leakage-prevention properties of polystyrene [115]. Furthermore, the specific gravity of vacuum-impregnated lightweight aggregate increased from 0.71 to 0.99 while the water absorption decreased from 67.05% to 2.19% [102].

Vacuum impregnation of lightweight aggregates is a new technique used to improve the properties of lightweight aggregates. The lightweight aggregate used in this technique is normally obtained from recycled waste such as recycled concrete aggregate or natural lightweight aggregates. The artificial aggregate-based geopolymer is type of porous aggregate and vacuum impregnation method that is an effective way to improve the properties by filling up the pores inside the artificial aggregates. However, different types of treated materials will provide distinct optimization designs. Thus, additional research should be conducted on various types of materials and their optimal conditions, including the mix design, relative humidity, and impregnation length. In current research, there is still no study on the lightweight aggregate with a geopolymer carried out by using this treatment, and this requires further investigation.

## 7. Conclusions and Summary of Future Work

In conclusion, the use of geopolymer in the manufacture of artificial aggregates has been shown to increase the properties and quality of the aggregates produced. Cold bonding technology demonstrated excellent aggregate properties in terms of strength while conserving energy since it does not require high temperatures in the curing and hardening process. However, the aggregates generated by sintering and autoclave processes always have high quality in terms of strength, water absorption, and the light weight. Incorporation of geopolymers also contributes to uniform pore distribution and can achieve lightweight aggregates at lower temperatures than those without the inclusion of geopolymers. Thus, artificial aggregates with geopolymer are believed to improve the properties of aggregates generated at lower temperatures, saving 20–50% of energy. There is a need for additional research on the performance of geopolymer-based artificial aggregates in diverse materials rich in Si and Al, as well as a comparison with artificial aggregates created without geopolymer.

Furthermore, the ITZ is an important determinant of the collective strength. The ITZ of artificial aggregate-based geopolymers showed better improvement in characteristics compared to natural aggregates used in concrete manufacturing. Strong ITZs in geopolymer concrete are associated with the existence of a layer of geopolymer paste with dense microstructures and a high concentration of the gel that binds aggregates and cement matrixes. The greater the ITZ of the aggregate and cement matrix, the higher the mechanical properties of the produced concrete. However, additional research is required to determine the features of ITZ in terms of thickness, chemical composition inside the ITZ, and the formation mechanism of ITZ between artificial aggregates and cement matrices.

Aside from that, a number of approaches to increase the attributes of high-quality artificial aggregates have been shown effective. Among the four techniques for improving aggregates, only soaking in an alkaline solution technique involves the incorporation of geopolymer in the current study to increase the aggregate’s strength and water absorption. However, the inclusion of a geopolymer via other techniques, such as two-step pelletization, coating, and vacuum impregnation, has not yet been addressed, making it a promising topic for future research. Future research could focus on the multiple components that act as geopolymer precursors as a treatment or process enhancement via two-step pelletization, coating, and vacuum impregnation. It is crucial to emphasise the mix design, relative humidity, correct viscosity, duration of the treatment procedure, and handling circumstances that impact aggregate quality, particularly when using the cold-bonding technique (not involving a high sintering temperature). This is due to the fact that the construction of artificial aggregates with geopolymer has had a positive impact, not only enhancing the aggregates’ quality, but also conserving energy at a lower temperature compared to aggregates generated without geopolymer.

## Figures and Tables

**Figure 1 materials-15-05516-f001:**
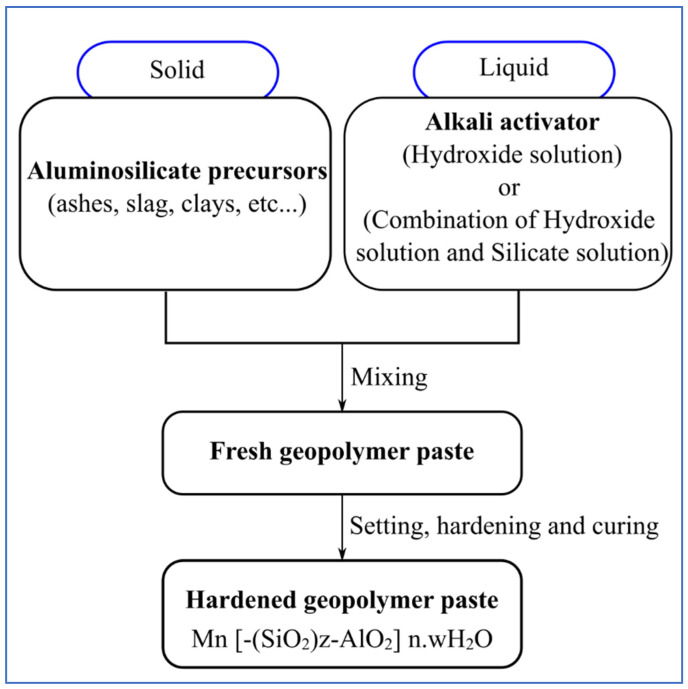
Illustration of Geopolymer Synthesis [47].

**Figure 2 materials-15-05516-f002:**
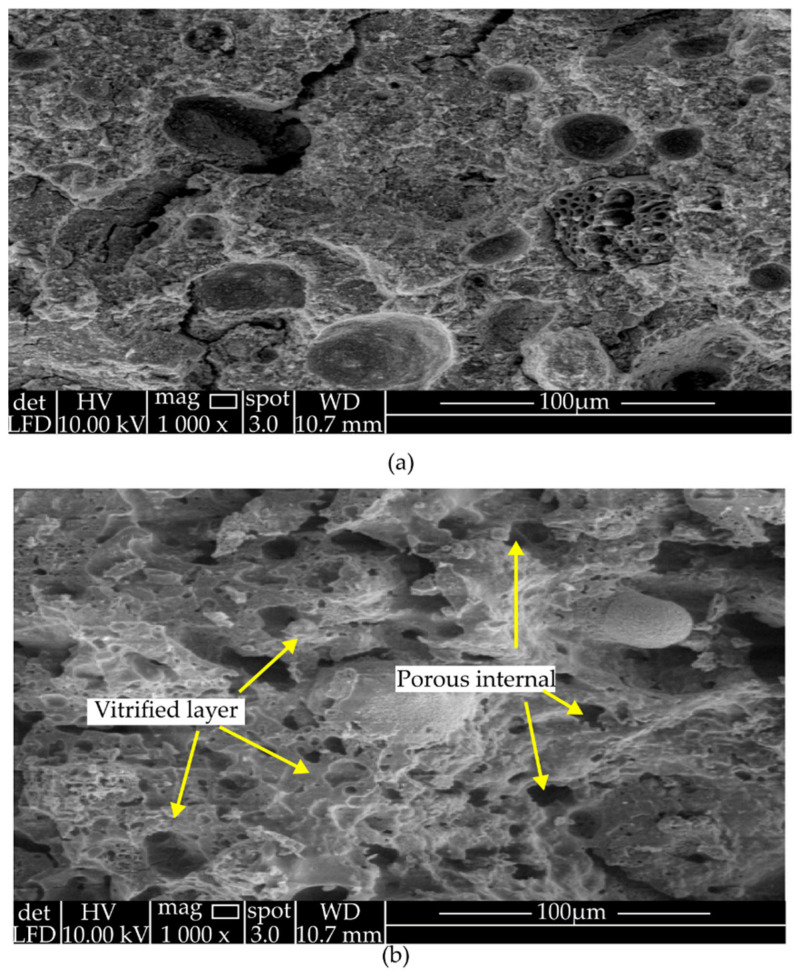
SEM micrograph of POFA aggregates (**a**) before sintering and (**b**) after sintering with magnification ×1000.

**Figure 3 materials-15-05516-f003:**
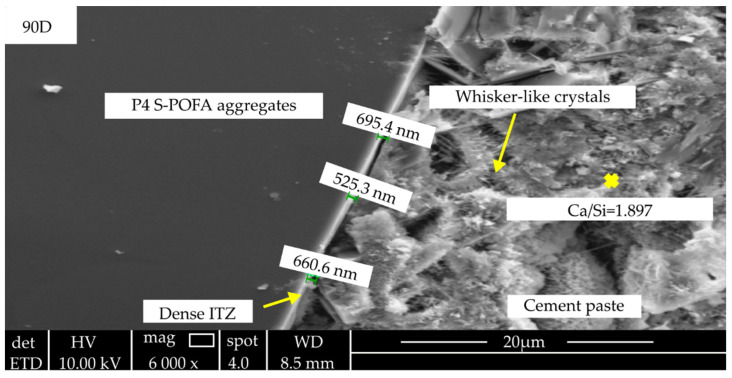
SEM images of S-POFA LWC ITZ (×6000) at 90 d [13].

**Figure 4 materials-15-05516-f004:**
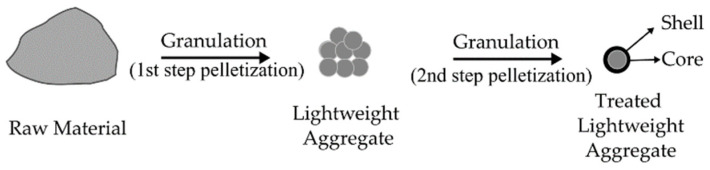
Two-step pelletization.

**Figure 5 materials-15-05516-f005:**
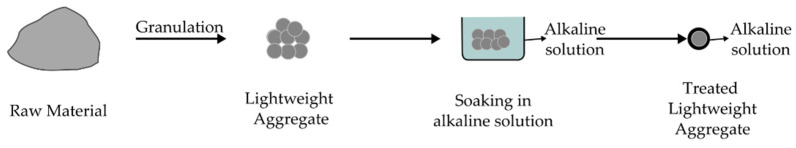
Soaking in alkali solution.

**Figure 6 materials-15-05516-f006:**
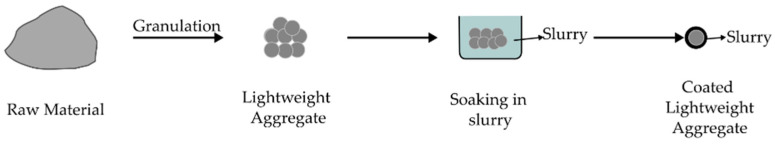
Coating.

**Figure 7 materials-15-05516-f007:**
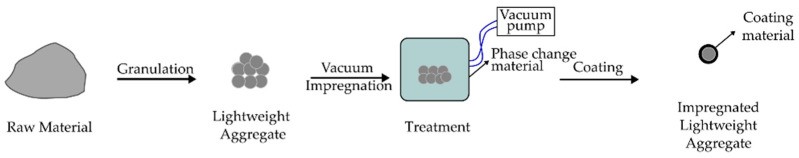
Vacuum impregnation.

**Table 1 materials-15-05516-t001:** Past research on Geopolymer-based Artificial Aggregates with different mixing parameter ratios and processing methods.

Researchers	Precursors		Addition of Additives	Mixing Parameter Ratio	Processing
	Raw Materials	Alkali Activator			
Tian et al. [37]	Red mud, coal fly ash	Na_2_SiO_3_	-	Solid-to-liquid ratio (0.34–0.39) Modulus ratio, Ms (1.29–3.84)	Shaping with disc pelletizer. Curing at temperature of 20 °C with relative humidity of 50%
Saleem et al. [38]	Class F fly ash, silica fume	NaOH, Na_2_SiO_3_	NaHCO_3_	Alkali activator percentage (NaOH-Na_2_SiO_3_) 20–80, 30–70, 40–60	Hand shaping and then microwave curing
UI Rehman et al. [12]	Coal fly ash and slag	NaOH, Na_2_SiO_3_	-	Different fly ash contents (80–90%), different slag contents (10–20%)	Cold-bond pelletization was applied. Dry curing with elevated temperature of 70 °C was used for curing aggregates
Parvathy et al. [39]	Class C fly ash, Class F fly ash	NaOH, Na_2_SiO_3_	-	The effect of different raw materials used on the performance of the fine aggregates (Class C fly ash and Class F fly ash)	Heating at 100 °C for 1 h and kept at ambient temperature for one day
Aslam et al. [40]	Class F fly ash and silica fume	NaOH, Na_2_SiO_3_	NaHCO_3_	Alkali activator ratio (NaOH:Na_2_SiO_3_) (0.42,0.53)	Hand shaping followed by microwave radiation curing.
Huynh et al. [34]	Class F fly ash and slag	NaOH, Na_2_SiO_3_	-	Different alkali equivalent, AE (5–9%) and alkali modulus, Ms (0.6–1.0)	Shaping by crushing the hardened paste of alkali-activated fly ash-slag divided into coarse and fine sizes by sieving
Wang et al. [41]	OPC, Class F fly ash, quarry tailings	NaOH, Na_2_SiO_3_	-	Effect of curing temperature (25–150 °C) and curing pressure (0.50–1.25 MPa)	Disk pelletization followed by autoclaving

NaHCO_3_ = Sodium carbonate, NaOH = Sodium hydroxide, Na_2_SiO_3_ = Sodium silicate.

**Table 2 materials-15-05516-t002:** Chemical composition of aluminosilicate precursors that have been used by previous researchers [53,54,55,56,57,58].

Chemical Composition	Class F FA	Kaolin	Metakaolin	GGBS	Red Mud	Rice Husk Ash
SiO_2_	54.40	48.10	55.57	32.00	22.82	87.40
Al_2_O_3_	32.10	36.90	41.55	14.10	15.06	3.00
CaO	1.10	0.20	-	44.22	12.24	1.40
K_2_O	0.20	1.90	0.43	0.31	1.19	0.49
MgO	0.80	0.17	0.05	5.32	0.27	-
Fe_2_O_3_	7.50	0.26	0.56	0.43	17.34	1.49
TiO_2_	2.10	0.25	0.26	0.62	3.43	-

Notes: FA = fly ash, GGBS = ground granulated blast furnace slag.

**Table 3 materials-15-05516-t003:** Findings on the utilization of cold bonding, sintering, and autoclaving as curing methods for geopolymer-based artificial aggregates.

Method	Raw Materials	Liquid	Significant Conclusion	References
Cold bonding	FA with the addition of GGBS and RHA	NaOH, Na_2_SiO_3_	The addition of GGBS and RHA in both binary and ternary blends improved the crushing strength of the aggregates produced	Bui et al. [82]
	High calcium FA with different percentages of the addition of OPC and Ca(OH)_2_	Water	An increase in OPC and calcium hydroxide enhanced the properties of fly ash aggregates including the density and strength performance.	Narattha et al. [16]
	FA	NaOH, Na_2_SiO_3_	The artificial aggregates were proven to have acceptable properties compared with the commercialized expanded clay aggregates with less energy consumption during manufacturing	Risdanareni et al. [84]
Sintering	POFA	NaOH, Na_2_SiO_3_	The application of S-POFA LWC with optimized sintered S-POFA aggregate demonstrated the feasibility of this material, as evidenced by the concrete’s physical and mechanical performance.	Kwek et al. [13]
	FA	Na_2_SiO_3_	Sintered FA aggregates showed a denser structure with smaller non-interconnected pores proven to have a low bulk density due to weight loss	Terzic et al. [15]
	High calcium bottom ash (denoted as WP), Low calcium bottom ash (denoted as NV)	NaOH	Proven to outperform commercial aggregates due to the spherical shape, which enhances the workability of concrete and the sorption properties.	Balapour et al. [14]
Autoclaving	FA and quarry tailings	NaOH, Na_2_SiO_3_	Based on reported properties, artificial aggregates lead to better use of the space and meet the environmental and economic needs of the commercial sector and are also capable of shortening the curing period through autoclave methods.	Wang et al. [19]
	FA and quarry tailings	Water	Artificial aggregates have the potential to partially replace CS in the production of concrete, hence reducing the consumption of non-renewable resources.	Wang et al. [41]

Notes: FA = fly ash, RHA = rice husk ash, POFA = palm oil fly ash and GGBS = ground granulated blast furnace slag.

**Table 4 materials-15-05516-t004:** Reported research on the physical and mechanical properties of artificial aggregates with the inclusion of geopolymers.

References	Artificial Aggregates Produced	Physical Properties			Mechanical Properties		
		Specific Gravity	Density, kg/m^3^	Water Absorption, %	Aggregates: Impact Value (AIV), %	Aggregates: Crushing Value (ACV), %	Crushing Strength, MPa
Tian et al. [37]	FA-RM aggregates	NR	1007–1132	9.80–12.10	NR	NR	1.46–6.18
Saleem et al. [38]	FA-SF aggregates	1.700	738	18.98	10.24	NR	2.03–12.00
UI Rehman et al. [12]	FA-GGBS aggregates	NR	764–878	18.73–28.30	25.00–39.00	NR	NR
Aslam et al. [40]	FA-SF aggregates	1.800	710	17.95	10.03	NR	3.34–4.54
Gomathi et al. [89]	FA-BT, FA-MK, FA-GGBS aggregates	1.68–1.89	848–983	13.01–21.26	31.96–50.47	NR	14.51–22.81
Sharath et al. [90]	FA-GGBS, FA-BL, FA-BT aggregates	2.000–2.200	NR	13.40–24.30	20.40–50.20	18.5–49.40	0.4–4.3
Risdanareni et al. [84]	FA aggregates	NR	1450–1500	22.00–23.00	NR	NR	NR
Kasi et al. [91]	FA aggregates	2.058	NR	7.07	28.31	23.96	NR
Parvathy et al. [39]	CFA, FFA aggregates	2.40–2.45	NR	5.51–6.05	NR	NR	NR

Notes: BT = Bentonite, BL = lime, CFA = Class C fly ash, FFA = Class F fly ash, FA = fly ash, GGBS = ground granulated blast furnace slag, MK = metakaolin, NR = not reported RM = red mud and SF = silica fume.

**Table 5 materials-15-05516-t005:** Past research on the improvement techniques on artificial lightweight aggregates.

Techniques	Researcher	Lightweight Aggregate	Treatment Material	Inclusion Geopolymer
Two-step pelletization	Yang et al. (2021) [106]	Soil	Fly ash	-
Two-step pelletization	Drissi et al. (2020) [107]	Cement	Calcium hydroxide, waste concrete powder, and paraffin powder	-
Two-step pelletization	Tajra et al. (2018) [108]	Fly ash	Cement	-
Two-step pelletization	Colangelo et al. (2015) [101]	Municipal solid waste incinerator fly ash	Cement, hydrated lime, coal fly ash	-
Soaking in alkali solution	Kalinowska-Wichrowska et al. (2022) [109]	Certyd	Sodium Hydroxide (NaOH)	Yes
Soaking in alkali solution	Venkata Suresh and Karthikeyan (2016) [110]	Class C fly ash	Sodium Hydroxide (NaOH)	Yes
Soaking in alkali solution	Gesoǧlu et al. (2007) [111]	Fly ash	Water glass (Na_2_O + Si_2_O)	Yes
Coating	Dixit and Pang (2022) [112]	Expanded clay aggregate	Cement with silica fume, cement with fly ash	-
Coating	Ye et al. (2022) [59]	Fly ash	Polypropylene (PP), linear low-density polyethylene	Yes
Coating	Pongsopha et al. (2021) [113]	Porous aggregate	Phase change material and silica fume	-
Coating	Tajra et al. (2019) [11]	Expanded perlite	Cement, fly ash, expanded perlite powder	-
Vacuum impregnation	Haider et al. (2022) [114]	Volcanic stone	Paraffin, epoxy resin, silica fume	-
Vacuum impregnation	Hasanabadi et al. (2021) [115]	Expanded perlite	Paraffin	-
Vacuum impregnation	Uthaichotirat et al. (2020) [102]	Recycled waste	Paraffin	-

## Data Availability

Not applicable.

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
