# Peer review of "Geopolymer-Based Artificial Aggregates: A Review on Methods of Producing, Properties, and Improving Techniques"

_materials, 2022, doi:10.3390/ma15165516_

Round 1

Reviewer 1 Report

1- Revise the abstract to have the core findings and the novelty of the study in a clearer form.

2- All references and citations should be EXACTLY based on the style adopted by IJST. 

3- More discussions and justifications are to be added to the results.

4- introduction :- ( line 141) artificial aggregates generated from geopolymers and geopolymerization methods are viable alternatives to natural aggregates. How can use artificial aggregates with alkali properties but natural aggregates are inert?????

5- Line 194 :- The use of geopolymers can be based on the Si/Al ratio, as the microstructure of geopolymers varies significantly depending on the Si/Al ratio. explain the impact of Si/Al ratio on producing geopolymers?

6- 5-( line 667) Furthermore, the ITZ of natural aggregate concrete indicated that a water film often forms around the  coarse particles due to bleeding and wetting effects. It results in a more porous micro- structure in the ITZ, which results in increased water content. As a result, the concrete 670 with geopolymer-based artificial aggregate did not show weak ITZs in the area of the  aggregates, why artificial aggregate did not show weak ITZs?????

Reviewer 2 Report

Dear Authors

It is difficult to evaluate a "Review Article" because the Authors' selection  of the references is very subjective and form the basis for an individual choice of issues under study. 

Presented study is well organised and composed. It is interesting to me form a perspective of civil engineer. I miss a little some comments on application of "Geopolymers-based Artificial Aggregates". The review is focused on "Methods of Producing, Properties and Improving Techniques", but it does not explain what is the final goal of presented technologies.

Unfortunately, as a Reviewer, I should not push the Authors towards my contributions not to my co-Authors' papers.

I believe that presented study may be published after some minor editorial corrections that I listed below:

0. Please reconsider the title. Id suggest "Geopolymer-based ..." and not "Geopolymers-based ..."

1. Please format the references according to MDPI template.

2. Please try to split cluster citations eg. [4-8]. Every cited paper deserves a cautious description in the introductory part of your study, just to let the Reader know of its importance and to prove the relevance of the citation.

3. Please check the relevance of [13,14] in the context in line 44.

4. Please check line 46. I suppose that you meant to refer here  to [13,14].

5. Please develop some comments in the Conclusions to point on the importance of presented technologies and to show prospects for practical applications.

Best regards

Reviewer 3 Report

This paper systematically reviewed the geopolymer-based artificial aggregate. In general, this is a very interesting paper which can provide new insights into the cement and concrete industry. I have some recommendations.

1. The common raw materials for geopolymer preparation and the application should be briefly reviewed. The recently published papers about  new precursor materials should be mentioned. For example, the waste glass powder-based geopolymer has been studied in recent years. ("A state-of-the-art review of crushed urban waste glass used in OPC and AAMs (geopolymer): Progress and challenges. Cleaner Materials, p.100083."). Some studies also mentioned the use of glass powder based geopolymer to stabilize the road bases.

2. The difference between geopolymer and alkali-activated materials should be clarified. These two materials have the similar preparation method, but people have started to view these two materials differently.

3. The sustainability and environmental concerns about the use of geopolymer should be mentioned. The leaching of free alkali may cause the pollution.

4. More discussion should be provided about the limitations of current studies and future work.

Round 2

Reviewer 3 Report

This paper has been revised based on the comments.